# Interpretable histopathology-based prediction of disease relevant features in Inflammatory Bowel Disease biopsies using weakly-supervised deep learning

Ricardo Mokhtari[1]                          RICARDO.MOKHTARI@ASTRAZENECA.COM
Azam Hamidinekoo[1]                        AZAM.HAMIDINEKOO@ASTRAZENECA.COM
Daniel Sutton[1]                              DANIEL.SUTTON@ASTRAZENECA.COM
Arthur Lewis[1]                                ARTHUR.LEWIS@ASTRAZENECA.COM
Bastian R. Angermann[2]                    BASTIAN.ANGERMANN@ASTRAZENECA.COM
Ulf Gehrmann[2]                              ULF.GEHRMANN@ASTRAZENECA.COM
Pål Lundin[3]                                  PAL.LUNDIN1@ASTRAZENECA.COM
Hibret Adissu[4]                              HIBRET.ADISSU@ASTRAZENECA.COM
Junmei Cairns[2]                              JUNMEI.CAIRNS@ASTRAZENECA.COM
Jessica Neisen[2]                              JESSICA.NEISEN@ASTRAZENECA.COM
Emon Khan[5]                                  EMON.KHAN@ASTRAZENECA.COM
Daniel Marks[3]                              DANIEL.MARKS@ASTRAZENECA.COM
Nia Khachapuridze[2,3]                      NIA.KHACHAPURIDZE@ASTRAZENECA.COM
Talha Qaiser[1]                                TALHA.QAISER1@ASTRAZENECA.COM
Nikolay Burlutskiy[1]                        NIKOLAY.BURLUTSKIY@ASTRAZENECA.COM

[1] *Imaging and Data Analytics, AstraZeneca, Cambridge, UK*

[2] *Translational Science and Experimental Medicine, AstraZeneca, Gothenburg, Sweden*

[3] *Early Clinical Development, AstraZeneca, Cambridge, UK*

[4] *Respiratory and Immunology Safety Pathology, AstraZeneca, Gaithersburg, USA*

[5] *Late Respiratory and Immunology, AstraZeneca, Cambridge, UK*

**Editors:** Accepted for publication at MIDL 2023

## Abstract

Crohn's Disease (CD) and Ulcerative Colitis (UC) are the two main Inflammatory Bowel Disease (IBD) types. We developed interpretable deep learning models to identify histological disease features for both CD and UC using only endoscopic labels. We explored fine-tuning and end-to-end training of two state-of-the-art self-supervised models for predicting three different endoscopic categories (i) CD vs UC (AUC=0.87), (ii) normal vs lesional (AUC=0.81), (iii) low vs high disease severity score (AUC=0.80). With the support of a pathologist, we explored the relationship between endoscopic labels, model predictions and histological evaluations qualitatively and quantitatively and identified cases where the pathologist's descriptions of inflammation were consistent with regions of high attention. In parallel, we used a model trained on the Colon Nuclei Identification and Counting (CoNIC) dataset to predict and explore 6 cell populations. We observed consistency between areas enriched with the predicted immune cells in biopsies and the pathologist's feedback on the attention maps. Finally, we identified several cell level features indicative of disease severity in CD and UC. These models can enhance our understanding about the pathology behind IBD and can shape our strategies for patient stratification in clinical trials.

**Keywords:** Weakly supervised learning, self-supervised learning, attention maps, IBD.

## 1. Introduction

Inflammatory bowel diseases (IBD) are chronic, relapsing-remitting inflammatory disorders of the gastrointestinal (GI) tract. Crohn's disease (CD) and Ulcerative colitis (UC) represent the two main types of IBD. Both result from a complex interplay of several factors, including abnormal immune responses, genetics, microbiome and environmental triggers (Baumgart and Carding, 2007). CD can affect any region of the gut, and is characterised by segmental mucosal ulceration, transmural inflammation, fissures, fibrosis, and stricture formation. In contrast, UC principally involves the colon, and manifests as continuous mucosal inflammation extending from the rectum proximally, with variable extent, and a more superficial inflammatory infiltrate. CD and UC share similar symptoms, but are pathophysiologically distinct diseases (Shanahan, 2001).

IBD is diagnosed based on a combination of clinical presentation, radiographic, endoscopic and histopathological findings (DeRoche et al., 2014). Defining the extent and severity of inflammation in IBD can influence treatment decisions and support prediction of a patient's prognosis. While endoscopic evaluation assesses the macroscopic tissue, histopathological evaluation assesses the microscopic tissue and is typically carried out through a trained pathologist's visual inspection of a Haematoxylin & Eosin (H&E)-stained tissue, digitised into a whole slide image (WSI). While there is a strong correlation between endoscopic and histopathological assessment (Irani et al., 2018), the relationship between these two data modalities (especially the relationship of endoscopic scores to histopathology) is not completely understood (Lemmens et al., 2013) and can potentially be improved through the development of interpretable machine learning models that predict endoscopic categories from H&E stained WSIs coupled with the interpretation of model predictions by a pathologist.

In this paper we demonstrate that applying self-supervised learning coupled with weakly-supervised learning to H&E-stained IBD biopsies can accurately distinguish disease type, macroscopic tissue appearance, and endoscopic scores. Specifically, we trained two recent state-of-the-art architectures, including Dual-Stream Multiple Instance Learning (DSMIL) (Li et al., 2021a) and Hierarchical Image Pyramid Transformer (HIPT) (Chen et al., 2022) on the large SPARC IBD dataset containing 1394 WSIs, and explored two training strategies - fine-tuning and end-to-end (E2E) training. We explored the relationship between endoscopic prediction and histology through pathologist collaboration, where we found that the high attention regions identified by the models were confirmed, qualitatively, to contain epithelial and stromal morphological/structural features consistent with inflammation. We further validated these models by leveraging a model trained on the publicly available Colon Nuclei Identification and Counting (CoNIC) dataset (Graham et al., 2021), which can segment and classify 6 types of cells. These predictions were compared with the attention maps produced by the weakly-supervised models.

These models have applications in clinical trials since they can speed up the workflow of pathologists by ranking WSIs by disease severity so that pathologists can prioritise the most severe tissues first as well as drawing their attention to the most diseased regions in the WSI. To our knowledge, this is the first work on exploring and validating weakly-supervised methods to associate endoscopic appearance with H&E morphology for large-scale IBD biopsies. Our code is available online at https://github.com/AstraZeneca/ibd-interpret.

## 2. Related work

WSI classification using only slide-level labels is commonly framed as a multiple instance learning (MIL) problem (Augustine, 2022) in which image patches are mapped to fixed-length embeddings followed by aggregation by an operator, for example max-pooling. However, these MIL methods suffer from several limitations that have been recently addressed by the research community. For example, MIL methods are prone to misclassifications in the case of an unbalanced number of positive instances when using a simple aggregation operation such as max-pooling (Li et al., 2021a). This can be addressed by utilising attention-based aggregation where each instance is given a specific attention weight during training (Tomita et al., 2019). Additionally, MIL methods can ignore the correlation among different instances across the WSI, which can be mitigated by using transformers to consider morphological and spatial information (Shao et al., 2021) as well as generating attention maps for improved interpretability. Attention map interpretability was further improved by incorporating non-local attention into a dual-stream MIL (DSMIL) architecture to calculate an attention weight for each patch (Li et al., 2021a). MIL models often use fixed patch-based features extracted by a CNN or only features extracted by a fine-tuned model since end-to-end training is expensive and time-demanding for large slides. These sub-optimal extracted features can be improved by incorporating a multi-resolution feature fusion mechanism to leverage varied-size tissue features such as glands vs cells (Li et al., 2021b). Recently, HIPT (Chen et al., 2022) extended multi-resolution feature fusion by introducing hierarchical pre-training, enabling a bottom-up aggregation of tissue features from cells to tissue morphology.

Following these advancements, we leverage DSMIL (Li et al., 2021a) and HIPT (Chen et al., 2022) due to their superior performance over the aforementioned methods on large public H&E classification datasets such as The Cancer Genome Atlas (TCGA). DSMIL achieves strong performance by combining self-supervised pre-training using SimCLR (Chen et al., 2020) with a dual-stream attention-based MIL aggregator. Since both local and global contexts are important for WSI classification, DSMIL takes patches extracted at multiple magnifications with the same resolution as input. This is disadvantageous since fine details are lost at low magnification. By contrast, HIPT extracts patches at a fixed high magnification but with a relatively large resolution, such that fine and coarse-grained details are captured at the same resolution.

In this paper, we predict clinically relevant IBD endoscopic categories including disease types, macroscopic appearances, and endoscopic scores from H&E biopsies which, to our knowledge, has not been addressed previously. Prior work mostly focused on electronic health records (Reddy et al., 2019), genomics, metagenomics (Abbas et al., 2019), biological and clinical parameters for predicting endoscopic scores in UC patients (Popa et al., 2020). Recently a fully supervised approach using H&E images to predict remission and the Nancy Histological Index scores for UC was investigated by Najdawi et al. (2022). In contrast, we consider both CD and UC and use weakly supervised methods with macroscopic patient level labels, without any detailed pathologist annotations at the microscopic level, which are inexpensive relative to the supervised method.

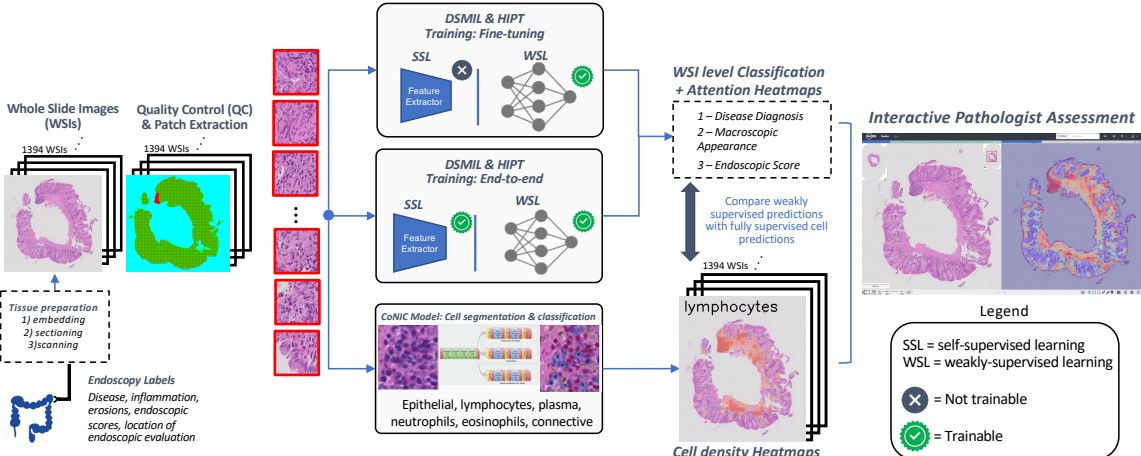

Figure 1: Overview of the implemented pipeline.

## 3. Methodology

The proposed methodology is summarised in Figure 1. First, we used an image quality control (QC) algorithm and extracted patches for training DSMIL and HIPT to predict 3 clinically relevant endoscopic categories and generate visual attention maps. A cell classification model was also applied to all slides and the individual cell predictions were aggregated into heatmaps for comparison with the weakly-supervised attention maps. Finally, attention maps and cell heatmaps were visualised in HALO Link for discussion with the pathologist. More details on these steps are in the subsections below.

### 3.1. Patch Extraction and Quality Control (QC)

To process gigapixel WSIs using deep learning models, we first split each WSI into many small, non-overlapping image patches using the Histolab (Marcolini et al., 2022) and CLAM (Lu et al., 2021) Python packages. We trained a QC model based on DenseNet (Huang et al., 2017) to automatically identify and remove regions from the WSIs with imaging and/or tissue artefacts such as overstaining, tissue folds, debris, out-of-focus areas and variations in contrast and hue markings. The QC model provided readouts at the slide-level, identifying how much of the slide was rejected as a proportion of the tissue. These readouts were used to exclude slides containing large areas of rejected tissue (e.g. >50%). The QC model was also integrated into the patch extraction pipeline to save individual patches containing >50% accepted tissue (see Appendix A).

### 3.2. Self-Supervised Pre-training and Weakly-Supervised Classification

In the following sections, we briefly outline how we distinguish self-supervised pre-training and weakly-supervised classification in DSMIL and HIPT - further details are provided in Section 4 and Appendix A.

**Self-supervised Pre-training.** We only consider the self-supervised representation learning components of DSMIL and HIPT. For DSMIL, this involves the SimCLR compo-

nent, while for HIPT, this involves the $ViT_{16} - 256$ and $ViT_{256} - 4096$ components, which learn to output embeddings at the patch level. These self-supervised components, once trained, are used to generate patch embeddings for all WSIs, which are subsequently used for training the weakly-supervised classification components.

**Weakly-supervised classification.** We only consider the weakly-supervised components of DSMIL and HIPT for predicting 3 endoscopic categories. For DSMIL the attention-based MIL aggregator, and for HIPT the $ViT_{4096} - WSI$ component were considered.

### 3.3. Interpretation of Model Predictions

Following weakly-supervised training, we generated visual attention maps, which facilitated optimal collaboration with the pathologist to interrogate the models' predictions.

**DSMIL attention maps** are visualised using the per-patch attention weights of MIL aggregator as a heatmap overlay on the WSI (Li et al., 2021a). A weight value close to 1.0 indicates that the patch contributes heavily to the final prediction compared with a patch that has a score close to 0.0. We generated these attention maps for all WSIs in our dataset.

**HIPT attention maps** are a natural part of HIPT due to its transformer backbone (Chen et al., 2022). The attention maps in the original paper demonstrated that they can highlight unique cancer-relevant tissue morphologies. Therefore, we also sought to understand what IBD-relevant tissue morphologies HIPT could learn.

**HALO Link Integration** was used to make the attention map review process interactive and straightforward for the pathologist. HALO is an image analysis platform for quantitative tissue analysis in digital pathology. We used HALO Link for sharing the WSIs and our predictions with the pathologists. The pathologist was not aware of any slide level endoscopic labels prior to assessment. For each slide, we asked the pathologist to describe the critical histopathological features for each slide including identification of inflammatory regions cellular composition, and morphological/structural features of the epithelium and stroma. We used HALO Link[1] to compare the attention maps and the pathologist comments side by side to interpret features learnt by the models.

A **cell prediction model** for detection and classification of six cell types trained on the publicly available annotated CoNIC H&E dataset (Graham et al., 2021) was used for understanding 6 cell populations. The CoNIC dataset was part of a grand medical challenge with the aim of predicting six types of cells on H&E slides, including epithelial cells, neutrophils, lymphocytes, plasma cells, connective cells, and eosinophils. We aggregated the cell predictions into statistics per patch and then visualised these as the heatmaps (Appendix E). The heatmaps were compared with the attention maps produced by DSMIL and HIPT and several representative examples were reviewed by the pathologist. Finally, several human interpretable features (HIFs) were calculated from the readouts and correlated with the endoscopic scores for both CD and UC - see Appendix E for an example calculation.

### 4. Experimental Setup

We included 1394 H&E-stained biopsies from 418 CD and 218 UC patients enrolled in a multi-centred longitudinal Study of a Prospective Adult Research Cohort with IBD (SPARC

---

1. https://indicalab.com/halolink/

IBD). The samples reside in the IBD Plexus database, provided with informed consent by the Crohn's and Colitis Foundation (Raffals et al., 2022). The total tissue area in 1394 biopsies is $\approx 7500 \mu m^2$ whereas the TCGA-lung resection dataset used to train DSMIL in the original work (Li et al., 2021a) has a total area of $\approx 65000 \mu m^2$ across 1054 slides, making SPARC IBD a relatively challenging dataset for classification.

We used 3 clinically-relevant labels from the SPARC IBD dataset that were acquired from endoscopy to define weakly-supervised classification tasks. These included: i) disease diagnosis (num. WSIs: CD=903 vs UC=491), ii) macroscopic tissue appearance (num. WSIs: normal=922 vs lesional=472), and iii) CD and UC-specific endoscopic severity scores (num. WSIs: low (CD=714, UC=134) vs high (CD=281, UC=131) score) - see Appendix A for more details on the exact splits. We used 4 NVIDIA A100 GPUs for training.

The QC model was trained on representative subset of 19 WSIs. We reserved a set of 10 WSIs for testing on which the Dice score of the QC model in segmenting background, good tissue and artefacts was 0.730. The QC model was then applied to all SPARC IBD WSIs. We excluded 11 slides with >50% rejected tissue and then DSMIL and HIPT were trained on patches extracted at 40x (Appendix A).

In training, we initially used DSMIL and HIPT models, which were pre-trained on the TCGA dataset and fine-tuned on our dataset, in order to assess the transferability of knowledge from TCGA cancer resections to IBD biopsies. In fine-tuning, the weights of the self-supervised components of both models (SimCLR in DSMIL and $ViT_{16}-256$ & $ViT_{256}-4096$ in HIPT) were frozen and only the weakly-supervised classification components (MIL aggregator in DSMIL and $ViT_{4096}-WSI$ in HIPT) were trained to predict the weakly-supervised classification tasks in SPARC IBD. We labelled the models that were fine-tuned on SPARC IBD as DSMIL-F and HIPT-F.

We also explored training DSMIL and HIPT end-to-end (E2E) on SPARC IBD. The self-supervised and weakly-supervised components of both models were trained on SPARC IBD from scratch to compare with the performance of fine-tuned models mentioned previously. We labeled the models that were trained end-to-end on SPARC IBD as DSMIL-E2E and HIPT-E2E (see Appendix A). In self-supervised and weakly-supervised training, we closely followed the training settings in the original papers (Li et al., 2021a; Chen et al., 2022).

In weakly-supervised training, we performed 5-fold cross validation, with an 80:20 split in all three prediction categories stratified on patients and ensuring that distributions of disease diagnosis, macroscopic appearance, biopsy location and the target label were consistent across all training and testing splits (more details in Appendix A). The same cross-validation splits were used for both DSMIL and HIPT models in all experiments. We used area under the receiver-operator characteristic curve (AUROC) to measure predictive performance. For further evaluation, the visual attention maps for all slides were generated and compared with the cell prediction heatmaps from the CoNIC model. To obtain a quantitative evaluation within reasonable pathologist effort, we sorted the WSIs by tissue size and selected 10 WSIs from CD and UC patients, each containing 5 normal and 5 lesional.

## 5. Results and Discussion

**Model Performance.** In Table 1 we compare the performance of end-to-end and fine-tuned DSMIL and HIPT models for predicting 3 weakly-supervised tasks in SPARC IBD.

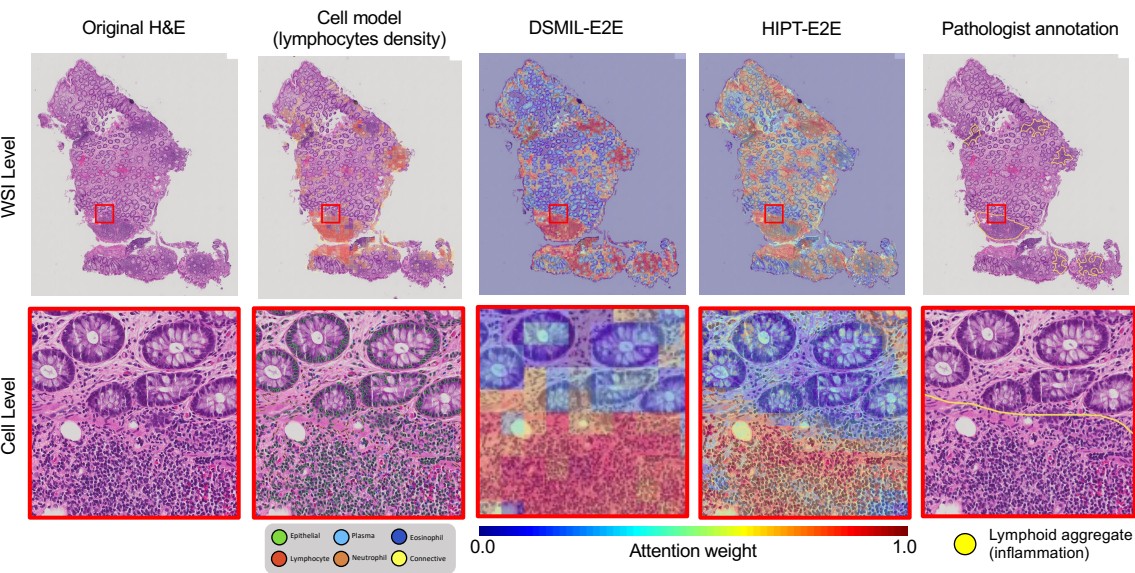

Figure 2: H&E image, Attention maps, cell level predictions and pathologist annotation for UC patient with lesional macroscopic appearance and high endoscopic score.

Table 1: Mean AUROC ± standard error (5-fold cross validation) of trained models across different prediction tasks.

| Model | Disease Diagnosis | Macroscopic Appearance | Endoscopic Score (CD) | Endoscopic Score (UC) |
|---|---|---|---|---|
| DSMIL-F | 0.656±0.007 | 0.522±0.008 | 0.528±0.016 | 0.592±0.007 |
| DSMIL-E2E | 0.692±0.010 | 0.750±0.006 | 0.740±0.009 | 0.634±0.017 |
| HIPT-F | 0.825±0.017 | 0.780±0.012 | 0.766±0.026 | 0.788±0.034 |
| **HIPT-E2E** | **0.865±0.019** | **0.814±0.008** | **0.786±0.017** | **0.802±0.014** |

Table 2: Performance of DSMIL-E2E and HIPT-E2E trained on clinically-relevant subsets of SPARC IBD for the task of predicting macroscopic appearance.

| Model | All data | Just CD | Just UC | Just Ileum | Just Colon |
|---|---|---|---|---|---|
| DSMIL-E2E | 0.750±0.006 | 0.689±0.010 | 0.808±0.013 | 0.586±0.010 | 0.775±0.019 |
| HIPT-E2E | 0.814±0.008 | 0.804±0.010 | 0.837±0.015 | 0.739±0.032 | 0.823±0.036 |

HIPT-E2E significantly outperforms DSMIL-E2E across all tasks (two-tailed t-test: diagnosis - $p<0.0001$, macroscopic appearance - $p<0.0005$, endoscopic scores (CD) - $p<0.05$ and (UC) - $p<0.0001$). It is likely that both the spatial patterns among cells and the context of the tissue microenvironment are well captured by HIPT's transformer backbone, leading to improved performance. In addition, we found that E2E training on SPARC IBD was

the optimal strategy for both DSMIL and HIPT training across all tasks. We suggest that the embeddings learned by E2E models through self-supervised pre-training on SPARC IBD were more useful for downstream prediction tasks than pre-trained embeddings from TCGA. This difference can be attributed to the differences between IBD and cancer morphologies as well as the difference between TCGA resections vs IBD biopsies.

**Pathologist Evaluation.** Through the pathologist evaluation of 20 WSIs, there were 8 TP, 7 TN, 4 FP and 1 FN when using the endoscopic labels whereas there were 9 TP, 7 TN, 3 FP and 1 FN when using the pathologist's evaluation as ground truth. Therefore, while not perfectly related, endoscopic and histological labels are strongly correlated, and the model was able to learn histologically relevant features via supervised training on endoscopic labels. Additionally, we obtained pathologist annotations of inflammatory regions for 1 WSI (Figure 2) and computed the Dice coefficient between the annotation and the thresholded attention map, obtaining 0.625, suggesting that inflammatory infiltrate was well localised in this WSI.

**Cell Prediction Model.** From the cell model's predictions, we found that the most indicative HIF in UC was the ratio of neutrophils to all cells in the tissue (p=0.0007), while in CD it was the ratio of eosinophils to all cells in the tissue (p=0.0002). These findings are reflected in Alhmoud et al. (2020). Qualitatively, there are similarities between the cell prediction maps and the pathologist's description of inflammation (for example, Figure 5). However, the quantitative performance at the cell level must be explored further. As this is a labor-intensive and time-consuming task for pathologists, we suggest combining weakly-supervised learning with supervised models is an alternative, more efficient, approach.

**Applications.** The trained models can be used to automatically rank biopsies by disease severity allowing pathologists to save time by prioritising more severe biopsies. This is important since the majority of biopsies received by pathologists are within normal limits. For biopsies outside of normal limits, the attention maps can be used for guiding the pathologist during microscopic evaluation of potentially malignant areas. We intend to quantitatively assess the speed up of pathologist workflow in future works. For clinical applications, we explored training DSMIL and HIPT using only CD or only UC biopsies as well as exclusively ileum or colon biopsies for predicting macroscopic appearance since disease features can present differently within these subsets (Table 2). We found that model performance can be improved by training on just UC and just colon subsets, while performance decreases when training on just ileum, suggesting that this is a more difficult task due to fewer ileum biopsies. We consider these types of modifications critical in applying these models to clinical trials and will be further explored in future works.

## 6. Conclusion and Future Work

We demonstrated that weakly supervised learning can produce accurate models trained on H&E-stained biopsies with endoscopic labels only. We observed potential in the publicly available CoNIC model and the attention maps. However, further quantitative and qualitative assessment of their generalisability is needed. Through reviewing the attention maps with pathologists, we can potentially better understand the relationship between histological and endoscopic labels. We plan to validate the trained SPARC IBD models with the corresponding attention maps on external IBD datasets from clincal trials.

## Acknowledgments

We would like to thank AstraZeneca for sponsoring and supporting this research and Crohn's & Colitis Foundation for providing the SPARC IBD dataset.

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

## Appendix A. Supplementary Methods

### A.1. QC and Patch Extraction

From applying our QC model to all 1394 WSIs, we found that 6, 11, and 40 slides had, respectively, >75, >50, and >25% rejected areas (Figure 3). Following QC, patches were extracted at 40x for training DSMIL-E2E (224 by 244 resolution) and HIPT-E2E (4096 by 4096 resolution), following the methods of (Li et al., 2021a) and (Chen et al., 2022). 2,054,901 patches were extracted for DSMIL and 13,952 patches were extracted for HIPT.

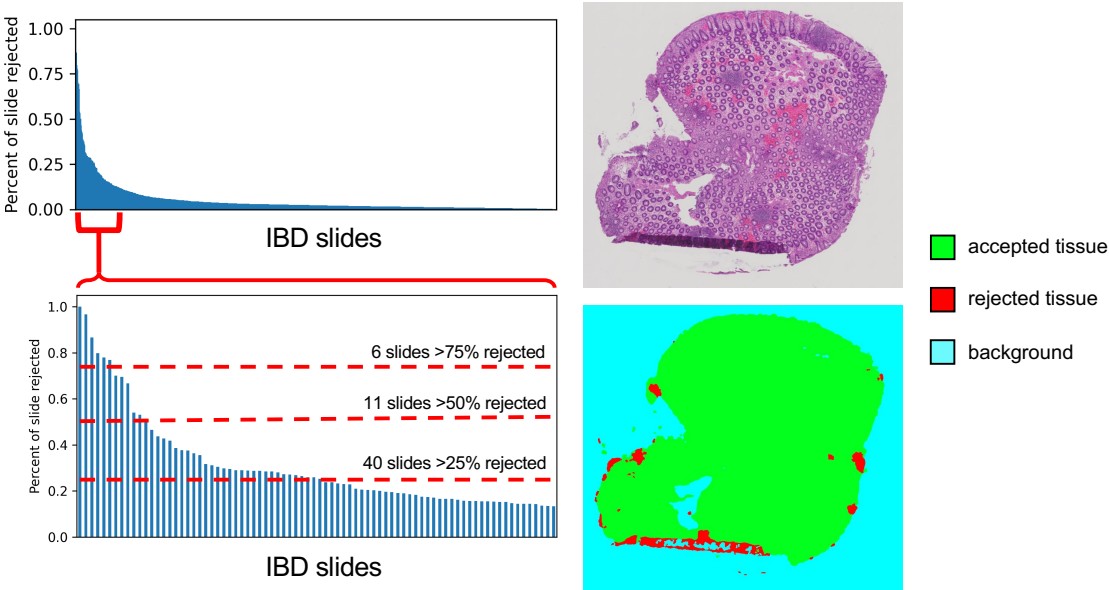

Figure 3: QC summary statistics on SPARC IBD and QC result on an additional slide

### A.2. Cross-Validation

5-fold cross validation is performed on in all weakly-supervised classification tasks in SPARC IBD. Splits are performed at the patient level, maintaining the distributions of biopsy location, disease diagnosis and the target label - see Figure 4 for the exact splits for all tasks. This was done to ensure that models are tested on a representative distribution of biopsies. The same cross validation splits are used for both DSMIL and HIPT experiments to allow them to be compared. For predicting macroscopic appearance, the "erosions/ulcers" and "inflammation" labels were combined into a "lesional" class. For endoscopic score, the median of SES score for CD and modified Mayo endoscopic score for UC were 0, so we treated the negative class as a score of 0 and positive class as score >0.

### A.3. Model Training

The hyperparameters used in self-supervised pretraining and weakly-supervised training for all DSMIL and HIPT models are shown in Table 3. To follow closely the methods of (Li

Figure 4: Proportion of patients (n=638) in original, train and test sets across 5 cross validation folds and across biopsy location, disease diagnosis and macroscopic appearance for all tasks. Proportions of patients in each category are kept consistent across all train and test splits.

et al., 2021a) and (Chen et al., 2022), we use the default parameters for both models across the board, but for HIPT we found that self-supervised pretraining converged after 30 epochs and in weakly-supervised learning we use only 1 transformer encoder layer. The same cross validation splits are used for both DSMIL and HIPT in all experiments.

Table 3: Hyperparameters used to train DSMIL and HIPT models.

| Model | Self-supervised pretraining | Weakly-supervised training |
|---|---|---|
| DSMIL-F | Frozen from TCGA | epochs: 200; embedding size: 512; learning rate: 0.0002; weight decay: 0.005 |
| DSMIL-E2E | epochs: 100; learning rate: 0.001; weight decay: 0.000001; batch size: 1024; arch: ResNet18 | epochs: 200; embedding size: 512; learning rate: 0.0002; weight decay: 0.005 |
| HIPT-F | Frozen from TCGA | epochs: 20; layers: 1; heads: 3; dropout: 0.25, learning rate: 0.0003 |
| HIPT-E2E | epochs: 30; weight decay: 0.04; learning rate: 0.0005, warmup epochs: 10 | epochs: 20; layers: 1; heads: 3; dropout: 0.25, learning rate: 0.0003 |

## Appendix B. Additional Experiments

Since DSMIL can take as input patches extracted at different magnifications, we assessed the performance of DSMIL in predicting disease diagnosis with different magnification datasets. These results are summarised in Table 4. In order to be consistent with HIPT we used 40x patches to train DSMIL-E2E for all main experiments (Table 1). However, we find that for predicting macroscopic appearance, slightly improved performance can be achieved with lower magnifications, although there is no statistically significant difference between these models trained at different magnifications and HIPT-E2E still significantly outperforms DSMIL-E2E at all magnifications.

Table 4: Comparison of DSMIL-E2E trained on different magnifications in predicting macroscopic appearance.

| Dataset | AUROC $\pm$ 1 SE |
|---|---|
| 5x | 0.645±0.008 |
| 10x | 0.752±0.006 |
| 20x | 0.758±0.006 |
| 40x | 0.750±0.006 |

# Appendix C. Additional Attention Maps and Pathologist Feedback

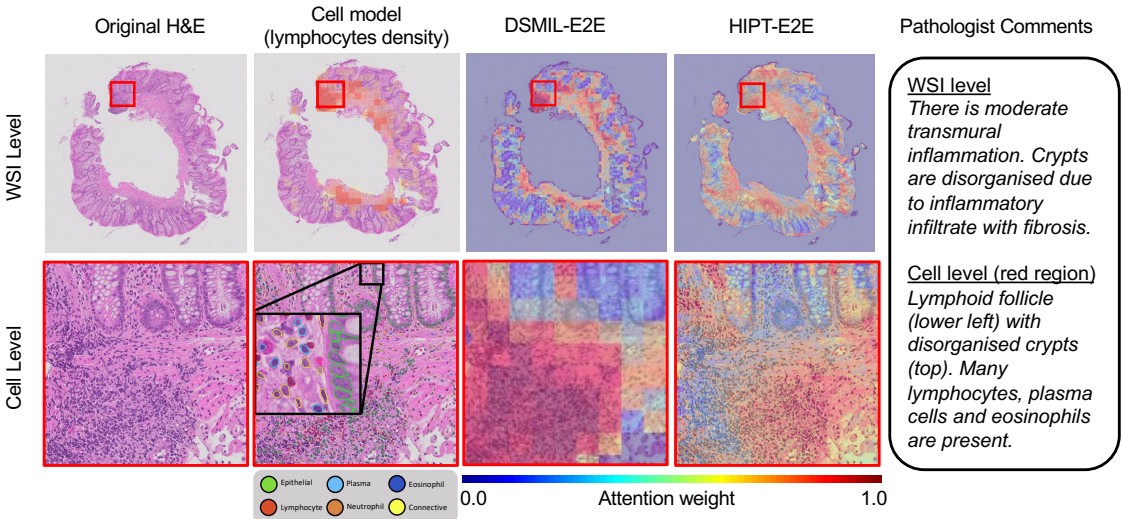

Figure 5: H&E image, lymphocytes density by the cell level model, attention maps from two models, at WSI level and cell level resolutions, and pathologist comments.

## Appendix D. Misclassified Cases

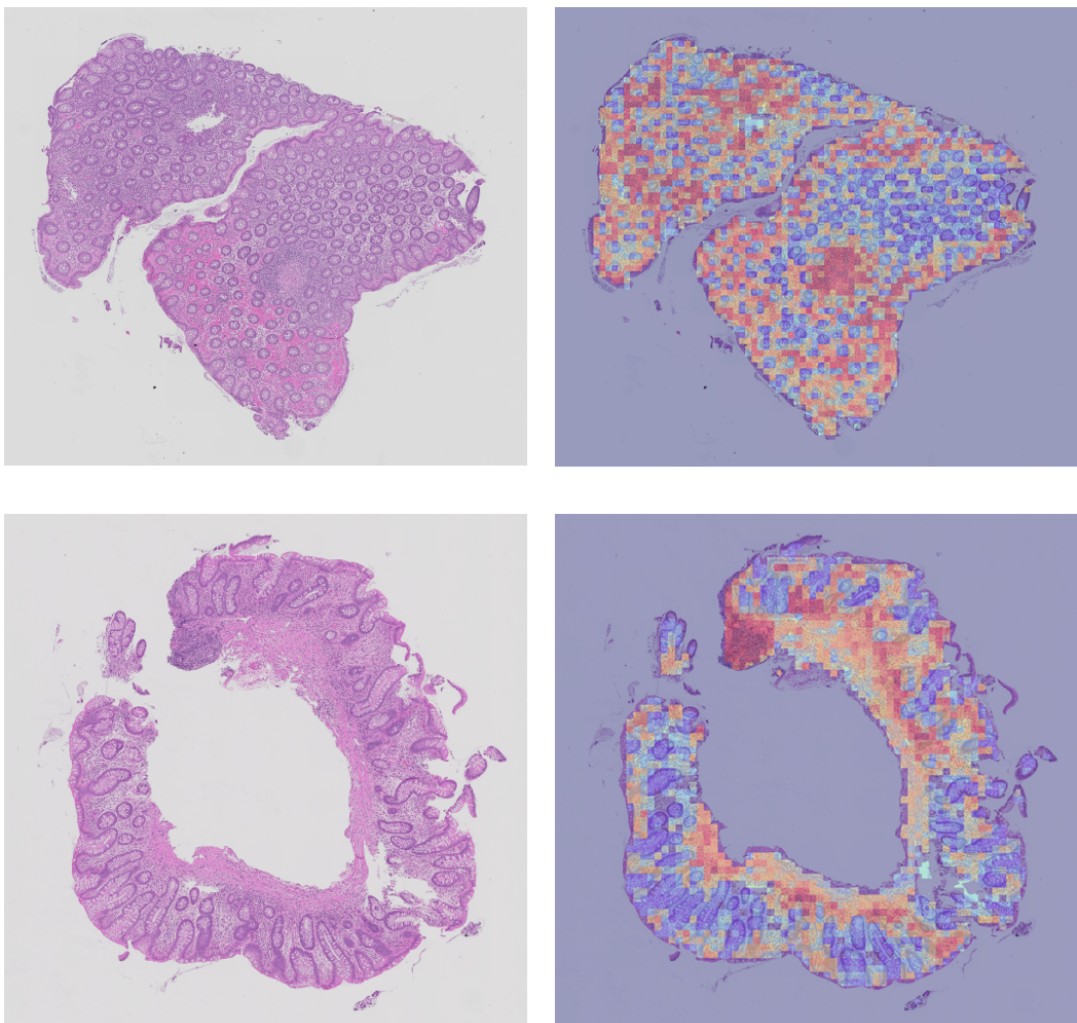

Figure 6: Two slides classified as "normal" from macroscopic appearance but predicted as "lesional" by DSMIl-E2E. Both slides were confirmed by the pathologist to contain inflammation and hence should be considered "lesional".

## Appendix E. Supervised Cell Model

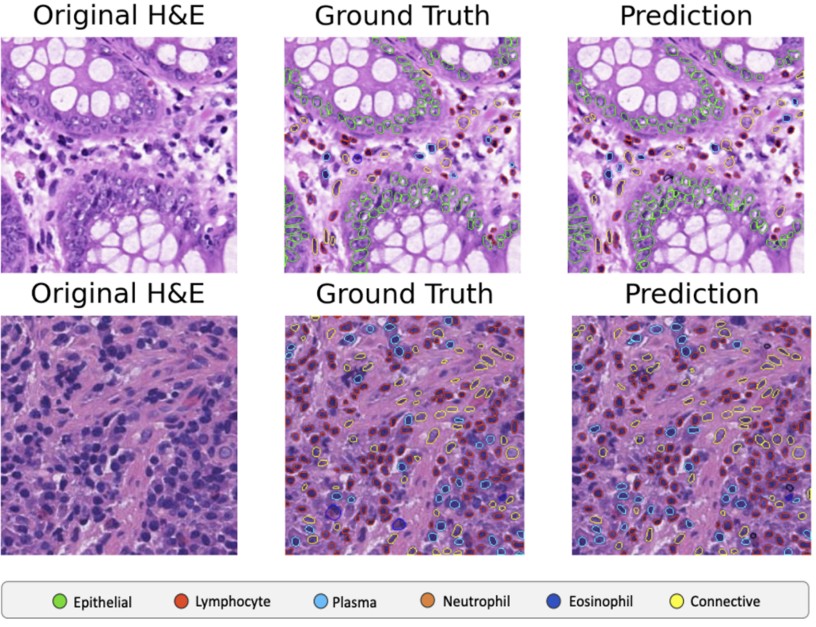

Figure 7: Six cell types predictions overlaid over two CoNIC patches.. CoNIC examples have ground truth, annotations by pathologists available in the dataset.

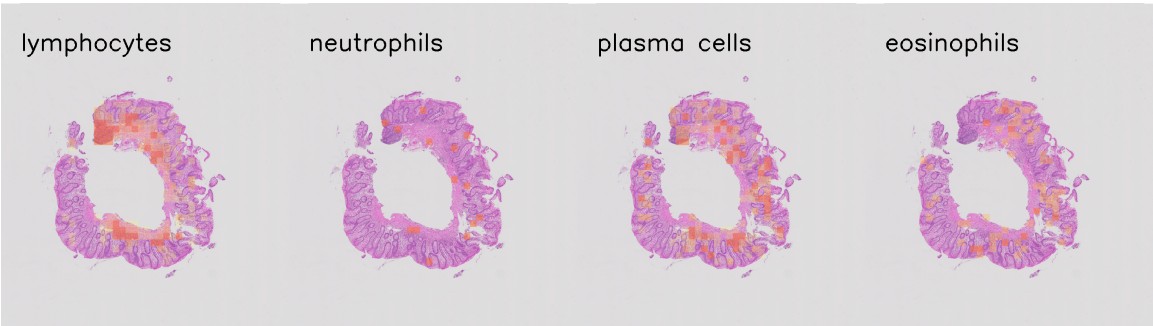

Figure 8: Four immune cell density heatmaps overlaid over an IBDplexus WSI.

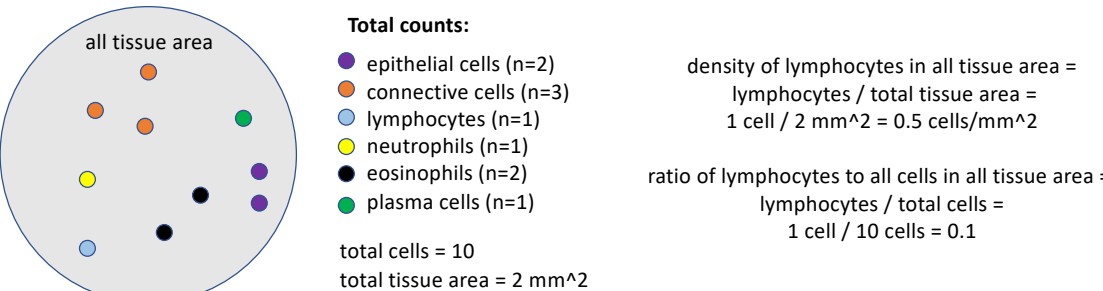

Figure 9: An example how human interpretable features (HIFs) are calculated from six class cells predictions.

