# OpenReview forum: "Interpretable histopathology-based prediction of disease relevant features in Inflammatory Bowel Disease biopsies using weakly-supervised deep learning"
_MIDL.io/2023/Conference — MIDL 2023 Poster_

### Official Review · Reviewer_7Zqg · 2023-02-03

**Confidence:** 3
**Preliminary Rating:** 4
**Recommendation:** Poster

**Summary:**

In this work, the authors present a solution for the automatic analysis of histopathologic images regarding IBD diagnostics in the absence of a reliable ground truth. The aim is to guide and assist physicians during histopathology towards severe cases and the most relevant regions.
To this end, the authors apply state-of-the-art, previously published methods to a dataset containing patients with Crohn's Disease (CD) and Ulcerative Colitis (UC) where weak, endoscopy-based labels are available, and propose a combination of self-supervised pertaining and weakly-supervised learning to still make use of this data.
Given the weak label information, the authors included a cross-verification via segmentation-based local statistics on cell types. Furthermore, they performed a validation study with a trained pathologist.

**Strengths:**

In general, the manuscript is written clearly and the authors did a good job in pressing a large pipeline with different settings plus several verification studies into an 8-page-paper without being too superficial.
I like the idea of making as much of the available data as possible, and highly appreciate the authors attempt to verify their findings since no reliable ground truth is available. A work with a clear focus on the application.

**Weaknesses:**

Unfortunately, I have the feeling that the scientific soundness of the work is not optimal. Since there is no methodological novelty (which is fine), the scientific impact lies on the translation of known concepts to the IBD scenario where the collection of a reliable ground truth is very challenging (as claimed in the introduction). Therefore, the authors use very coarse/unreliable labels taken from endoscopy data.
If I understood everything right, the prediction of these labels is just a proxy task since these labels are typically known before histopathologic images are available in a clinical setting. Thus, the presented quantitative results are a verification that the patterns th model extracted correlate to the right things. However, the useful thing to support the physicians here are the attention maps for guidance towards relevant regions and, thus, a potentially significant speed-up.
To be fair, the authors took the right approaches to evaluate these qualitative findings (cross-evaluation with cell detection, validation by human expert). However, it is these key evaluations that are described quite vaguely and without any kind of quantification! Since this is exactly the scientific value of this work (in my opinion), the authors should make this more concise and reliable!











**Deanonymize Review:**

yes

**Detailed Comments:**

Accordingly, I have some major concerns that should be addressed:
1) In section 5 "Application", the authors claim that "The trained models can be used to automatically rank biopsies by disease severity allowing pathologists to save time by prioritising biopsies with higher disease activity." From my understanding, the endoscopic labels are known before the histopathology is performed (assuming that the biopsies are taken during endoscopy). Thus, the disease severity should be already known, right? And in case this is not right and the relationship between severity and the endoscopic labels is vague, what makes you think that they are still sufficient for training a severity prediction tool (even in a weakly supervised way)? Please clarify!
1) Since the output of the cell-based prediction model is crucial for the evaluation, and the model is used as a tool, its performance should be quantified somewhere. The authors claim in section 5 that "The cell-based model correctly identifies multiple cell types at the WSI level, which we confirmed with the pathologist for several WSIs". What does "correctly" means? 100% accuracy across all cells? How many WSIs are "several"? Was it one pathologist? How were the WSIs that were validated manually selected?
2) The same holds for basically all steps of human validation. These results should be quantified and not only given vaguely (e.g. section 5 Attention Maps: "With the pathologist’s support, we confirmed numerous cases where both DSMIL-E2E and HIPT-E2E accurately and precisely localised areas of inflammation in the tissue"). Thus, a few follow-up questions:
2.1) How many pathologists served as observers? If more than one, how was the unity of opinion assessed?
2.2) What exactly did the pathologist see (or explicitly did not see) during which validation? E.g. section 5 Attention Maps, it is not clear whether the pathologist annotated data and these annotations were compared to the attention maps or whether the pathologist rated the confidence in the correctness of attention maps he or she was presented (and if that was the case, whether this was binary or using some kind of score). Both approaches might result in different sources of bias, and thus it is extremely important to have that information!
2.3) A quantification is crucial in the results of section 5 "Learning Histopathological Labels with Endoscopy Data". There are several vague descriptions like "identified several cases" or that there are slices where the findings of the model and the pathologist differed. Please provide statistics for these claims! I am a scientist, so I do not want to trust in authors but in data ;-)

Additionally, I have some minor concerns:
1.) The authors state that the QC model was trained on 19 WSIs. How did you select these images? And why did you choose to use a small number only? Since you use this network simply as a tool to achieve your goal and fully rely on the previously published study regarding its performance, why do you actively make the task harder? In the end, you want a model that predicts everything within your dataset and not something that generalizes across everything...
2.) The motivation could be more clear in the introduction. The text is very well written and a great overview of what you did, but I was not getting the clinical benefit of predicting endoscopy labels from WSI until I read the very last bit of the paper.


There are also some typos to fix:
1.) Page 2, line 11: Should start a new sentence here (?)
2.) HALO Link reference: In the MIDL citation format, this reference looks a bit weird (Labs, 2022). Maybe change the Bibtex file or cite it in a footnote?
3.) The abbreviation E2E suddenly appears on page 6 and is later used in the results. Please introduce it correctly somewhere.
4.) Appendix Figure 9: Delete one of the "feature"

**Paper Type:**

validation/application paper

**Questions To Address In The Rebuttal:**

My main concern is around the scientific soundness of the pathologists validation. As mentioned, I think that this is the scientific backbone of the paper, and thus its impact is even higher than the presented prediction results. Therefore, from a point of scientific rigor, I cannot vote for accepting this manuscript in its current form.

Please note that I really like the study itself and appreciate the effort you have taken, I actually believe it is exactly the right way to do it! Just give more soundness to the validation.

Please address my concern listed above.
I am sure it is possible to add some statistics and numbers within the rebuttal period. If the human validation is reliable and the description sufficiently accurate, I am happy to adapt my vote!

---

### Official Review · Reviewer_6chW · 2023-02-05

**Confidence:** 4
**Preliminary Rating:** 3
**Recommendation:** Poster

**Summary:**

While reading the paper from the start, it was hard for me to understand what the key contributions were. When reaching Table 1, it became clear that Table 1 is the key result of the paper. The authors used two recently proposed methods, DSMIL and HIPT, in two training settings, fine-tuning and full end-to-end training, and report results for 4 classification tasks that I found not clearly described except for the first one, in 5-fold CV, applied to 1397 WSIs from 418 CD and 218 UC patients. End-to-end training works much better than fine-tuning, HIPT works much better than DSMIL. I assume the authors considered this not interesting enough, so they added other experiments and include figures with feedback from a pathologist, but these additional results are not well fleshed out and seem a bit preliminary to me. I get the impression they only had one short session with the pathologist. There are many results in an appendix, but the idea of MIDL is that you present your main results in 8 pages.



**Strengths:**

Application addressed is interesting and important
Fairly large dataset (with only image level labels)
Good comparison between 4 approaches
Interesting ideas wrt final applications, such as sorting slides to be reviewed by the pathologist

**Weaknesses:**

Experiments are done on proprietary data?
No comparison with human reading
No code is shared
The experiments and data used to obtain Table 1 are not properly described and discussed
Unclear what we learn from additional experiments, eg Table 2, and the comparison with a pathologist, this part seems preliminary



**Deanonymize Review:**

yes

**Detailed Comments:**

Code should be shared. Without this code, the value of the paper is much lower. Research from Umeå University showed (https://arxiv.org/abs/2210.11146) that (MIDL) papers regularly promise to share code but ultimately the code is not shared or the code shared is not of high-quality. In my opinion, code should be shared with reviewers during the review period and conferences should require that code is really shared.

SPARC IBD dataset is from TCGA? Is it publicly accessible? Please provide a link. List number of images/patches, per label to provide a proper description of the dataset, explain training/validation subdivides. If the data is not public, the work is not reproducible.

Fig 1 not clear. Which predictions are made, what is eg location, which scores? Do you analyze 1 slide? What is QC processing? Is cell segmentation and classification using a conic algorithm or your contribution? When reading on I start to understand, but an overview Figure should be understandable on its own.

Embeddings at cell and patch level, later called patch embeddings? Confusing.

First pages not clear which dataset was used, how many slides were processed, how many cases were reviewed by the pathologist

HALO software is not clearly described. Is it public? It just seems a viewer?

HIFs, I don’t know what that is supposed to mean. In the main text you introduce concepts and definitions which are only later presented in the appendices. That should not be done. In general, appendices are too long and introduce new results, it seems appendices were used because you could not fit everything in the main paper.

Typo fine-tunning. Section 4 reads as a short diary, describe experimental setup and results properly. Make clear how many slides were used for each results in Table 1 and 2, how many positive/negative samples were there for each experiments.

Suggested application is to rank biopsies prior to pathologist’ review. Interesting idea, this could be analyzed, can the best algorithm produce a useful ranking? It is now more a suggestion for future work?

I do not understand what the task is for which Table 2 shows results. For somebody not very familiar with this application area, the paper is a bit of a puzzle at some places.

The main weakness is that I have no idea if the best results in Table 1/2 are good and pertain to a relevant task. If an average pathologist would achieve 0.95 my conclusion would be that using a weakly labelling and patch MIL approach for this task does not work. I suggest to set aside a test set and compare to a panel of pathologists, share your data, share you code as a baseline algorithm, and organize a challenge.

Not clear if HIPT and DSMIL are open source. If you share all your code in a repository with a permissive license it would be clear.

Why is HIPT so much better than DSMIL? Discuss this, preferably with experiments. Does this mean we should from now on only use HIPT? That would be a good contribution.

The motivation as stated in the introduction is vague, I get the idea the authors do not have annotated data and therefore decided to use this method. Would science not progress more if we collected annotated data? The introduction talks about developing tools to decode, this was a bit vague to me.

WSI classification based on weakly labeled biopsies? How many WSIs in a biopsy? Or is it single WSI classification?

TCGA has many collections, state specifically which one you used.



**Paper Type:**

methodological development

**Questions To Address In The Rebuttal:**

Please address all questions above.
--------------------------------------------------------------------------------------------------------------------------------------------------------------------

---

### Official Review · Reviewer_s8Qa · 2023-02-06

**Confidence:** 4
**Preliminary Rating:** 4
**Recommendation:** Oral

**Summary:**

In this paper, two recently published self-supervised methods are used to predict endoscopic categories from WSI images. Endoscopic categories are used as weak labels to train the deep learning models in a weakly supervised setting. Quantitative results show that end-to-end learning using the HIPT model performed best. In addition, the paper describes that qualitative inspection of the result using visual attention maps showed a strong association with histopathological inflammatory features of the disease. The authors conclude the paper with the claim that they showed that weakly supervised training can be used to produce accurate models for prediction of H&E slides for IBD.

**Strengths:**

- State-of-the-art methodology, good comparison of fine-tuning and end-to-end training
- Relatively large dataset, and good use of available public dataset
- Important addition to add the visual attention maps
- Well written and well explained paper

**Weaknesses:**

- The qualitative assessment of the results by the pathologist is mostly anecdotal and no systematic observer experiment is performed. Therefore, the claim that the visual attention maps show a strong association with inflammatory features is not confirmed by actual numbers. Therefore, I think this claim is currently too strong, and the authors should refer to future qualitative experiments to confirm this.
- No external validation of the method on a external validation set. Only 5-fold cross-validation is performed. Please add this as a limitation.
- I would have liked to see an analysis whether the resulting models differed across the 5 different folds in CV. Which parameters of the models were optimized per CV split, and did you find different parameters to be optimal across different CV splits? Or were most hyperparameters optimized over all 5 CV splits?


**Deanonymize Review:**

no

**Detailed Comments:**

See previous points.

**Paper Type:**

validation/application paper

**Questions To Address In The Rebuttal:**

- Please add how a good qualitative experiment to confirm the assumed association between visual attention maps and histopathological inflammatory features of IBD should be setup.
- Please add a discussion paragraph about the need for an external validation set

---

### Meta-Review · Area_Chair_6qiw · 2023-02-19

**Recommendation:** Accept (Poster)
**Confidence:** 5

**Metareview:**

Authors have presented a very interesting approach to leverage interpretable models to identify  features that are relevant to crohn's disease and ulcerative colitis. Reviewers have had a positive response in general. Despite the fact that there were some reservations, the revised manuscript has been very useful in clarifying many underlying doubts. The end result is all reviewers agree about the paper's importance and its potential to generate interest in MIDL community.